

# The suitability of atmospheric oxygen measurements to constrain Western European fossil-fuel $CO_2$ emissions and their trends

Christian Rödenbeck[1], Karina E. Adcock[2], Markus Eritt[1], Maksym Gachkivskyi[3], Christoph Gerbig[1], Samuel Hammer[3], Armin Jordan[1], Ralph F. Keeling[4], Ingeborg Levin[3], Fabian Maier[3], Andrew C. Manning[2], Heiko Moossen[1], Saqr Munassar[1], Penelope A. Pickers[2], Michael Rothe[1], Yasunori Tohjima[5], and Sönke Zaehle[1]

[1]Max Planck Institute for Biogeochemistry, Jena, Germany
[2]Centre for Ocean and Atmospheric Sciences, School of Environmental Sciences, University of East Anglia, Norwich, UK
[3]Institute of Environmental Physics, Heidelberg University, Germany
[4]Scripps Institution of Oceanography, University of California, San Diego, USA
[5]Center for Environmental Measurement and Analysis, National Institute for Environmental Studies, Tsukuba, Japan

**Correspondence:** C. Rödenbeck (christian.roedenbeck@bgc-jena.mpg.de)

**Abstract.** Atmospheric measurements of the $O_2/N_2$ ratio and the $CO_2$ mole fraction (combined into the conceptual tracer "Atmospheric Potential Oxygen", APO) over continents have been proposed as a constraint on $CO_2$ emissions from fossil-fuel burning. Here we assess the suitability of such APO data to constrain anthropogenic $CO_2$ emissions in Western Europe, with particular focus on their decadal trends. We use an inversion of atmospheric transport to estimate spatially and temporally

explicit scaling factors on a bottom-up fossil-fuel emissions inventory. Based on the small number of currently available observational records, our $CO_2$ emissions estimates show relatively large apparent year-to-year variations, exceeding the expected uncertainty of the bottom-up inventory and precluding the calculation of statistically significant trends. We were not able to trace the apparent year-to-year variations back to particular properties of the APO data. Inversion of synthetic APO data, however, confirms that data information content and degrees of freedom are sufficient to successfully correct a counterfactual prior.

Larger sets of measurement stations, such as the recently started APO observations from the Integrated Carbon Observation System (ICOS) European research infrastructure, improve the constraint and may ameliorate possible problems with local signals or with measurement or model errors at the stations. We further tested the impact of uncertainties in the $O_2:CO_2$ stoichiometries of fossil-fuel burning and land biospheric exchange and found they are not fundamental obstacles to estimating decadal trends in fossil-fuel $CO_2$ emissions, though further work on fossil-fuel $O_2:CO_2$ stoichiometries seems necessary.

## 1 Introduction

Anthropogenic carbon dioxide ($CO_2$) emissions from the burning of fossil fuels (coal, petroleum, natural gas) and from cement production are the primary cause of the rising $CO_2$ burden in the atmosphere causing recent climate change. Reducing these $CO_2$ emissions has therefore become an important political target. While the emissions are still rising globally from each year to the next, they already decreased in 24 countries during the decade 2012-2021, including in the European Union (Friedlingstein





et al., 2022). The ability to trace such decadal trends in $CO_2$ emissions is a pre-requisite to gain confidence in the effectiveness
of any political reduction measures.

Fossil-fuel $CO_2$ emissions are typically quantified by inventories based on energy or economic data combined with emission
factors (e.g., CDIAC (Andres et al., 2016), EDGAR (Janssens-Maenhout et al., 2019), TNO (Denier van der Gon et al.,
2017), GridFED (Jones et al., 2022)). Even though such a quantification is generally considered quite accurate, it is inherently
complex due to the variety of combustion processes and fuel types. Existing differences between the various inventories reflect
uncertainties, e.g. from possible omission or double-counting of contributions. Further uncertainty is related to the input data,
including emission factors and national emissions totals based on self-reporting by industries or individual countries. Andres
et al. (2014) report an uncertainty of 8.4% ($2\sigma$) for the global fossil-fuel $CO_2$ emissions in the CDIAC inventory. In an
uncertainty analysis for the TNO estimates, Super et al. (2020) found an uncertainty of 1% for the total of all considered $CO_2$
emissions within their study area comprising several highly industrialized European countries; however, this does not include
uncertainties due to "incompleteness of the emission inventory (i.e. if sources are missing) or double-counting errors." When
disaggregating larger-scale emissions totals (primarily known on the country level, e.g. from energy use statistics) onto smaller
spatial and temporal scales according to chosen proxy variables, uncertainties can increase considerably (Peylin et al., 2011;
Super et al., 2020). Decadal trends in the emissions, particularly relevant with regard to emissions reduction targets, might be
affected by specific uncertainties depending on whether the trends in all input quantities have been considered correctly. For all
these reasons, independent validation of emissions inventories is highly desirable and recommended by the "2019 Refinement
to the 2006 IPCC Guidelines for National Greenhouse Inventories" (Calvo Buendia et al., 2019).

Atmospheric measurements of oxygen ($O_2$) and carbon dioxide ($CO_2$) over continents (combined into the conceptual tracer
APO, see Sect. 2.1 below) have been proposed as a constraint on $CO_2$ emissions from fossil-fuel (FF) burning (Pickers et al.,
2022). As a proof of the concept, the relative reduction in fossil-fuel emissions due to the COVID-19 lockdowns in 2020 could
clearly be identified in the APO record from the WAO (Weybourne Atmospheric Observatory, East Anglia, UK). In order to
use the APO signals to infer absolute FF $CO_2$ emissions over a given region, however, the atmospheric transport from the
locations of the emissions to the measurement locations needs to be taken into account quantitatively. A technique widely used
for this purpose in the context of greenhouse gases (e.g. $CO_2$, $CH_4$) or air pollutants is the "atmospheric transport inversion"
pioneered by Newsam and Enting (1988). It has also been applied to APO (Rödenbeck et al., 2008), though so far with a
focus on estimating interannual variations in the sea–air oxygen exchange in order to diagnose variability in ocean-internal
processes relevant to oceanic $CO_2$ exchange. In that context, the flux estimation was constrained from APO observations at
remote marine locations, and the fossil-fuel emissions were prescribed as given by an inventory.

However, APO has also been measured at several locations on the European continent for more than a decade (Fig. 1,
Fig. 2). In this study, we consider whether and how these data can be used to quantitatively assess fossil-fuel $CO_2$ emissions
in Europe. We particularly focus on decadal trends in the fossil-fuel $CO_2$ emissions on subcontinental spatial scales, as both
being relevant in the context of emissions reduction targets and expected to be accessible to constraint by atmospheric data.
Specifically, we present first FF $CO_2$ emissions estimates for Europe (focusing on a Western part encompassing the UK,
France, The Netherlands, Belgium, Switzerland and parts of western Germany, see magenta color in Fig. 1) based on available





European APO time series (Sect. 3.1), test the potential information content of available or hypothetical sets of measurement stations (Sect. 3.2), and assess the magnitude of influences from uncertainty in the $O_2$:$CO_2$ stoichiometry of FF burning (Sect. 3.3) and of $CO_2$ and $O_2$ exchanges by the land biosphere (Sect. 3.4). On the basis of these results, we discuss the current situation and possible ways forward (Sect. 4).

## 2 Method

### 60  2.1  The APO flux – definition and contributions

The original definition of APO as a conceptual atmospheric tracer by Stephens et al. (1998) combines measurements of the atmospheric $O_2$ and $CO_2$ abundances. To very good approximation, this definition can be translated into the notion of a surface–atmosphere APO flux

$$f^{APO} = f^{O_2} + 1.1 \cdot f^{CO_2} - \frac{X_0^{O_2}}{X_0^{N_2}} \cdot f^{N_2} \tag{1}$$

(Appendix A of Rödenbeck et al., 2008). Here, $f^{O_2}$ and $f^{CO_2}$ are the surface–atmosphere fluxes of $O_2$ and $CO_2$, respectively. The last term of Eq. (1) represents a small contribution proportional to the surface–atmosphere nitrogen flux (involving the reference mole fractions $X_0$ of $O_2$ and $N_2$ in air), arising because the atmospheric $O_2$ abundance is reported as a molar ratio of $O_2$ to $N_2$. The only relevant $N_2$ flux originates from the ocean due to solubility variations with temperature.

The total surface–atmosphere $O_2$ and $CO_2$ fluxes comprise contributions by fossil-fuel burning ($f_{FF}^{O_2}$, $f_{FF}^{CO_2}$), the terrestrial 70 biosphere ($f_{NEE}^{O_2}$, $f_{NEE}^{CO_2}$ = NEE [net ecosystem exchange]), and the ocean ($f_{oc}^{O_2}$, $f_{oc}^{CO_2}$). The contributions by fossil-fuel burning and by the terrestrial biosphere occur in stoichiometric proportions denoted by

$$\alpha_{FF} = f_{FF}^{O_2} / f_{FF}^{CO_2} \tag{2}$$

and

$$\alpha_{NEE} = f_{NEE}^{O_2} / f_{NEE}^{CO_2} \tag{3}$$

respectively. Their contributions to the total APO flux thus are

$$f_{FF}^{APO} = (\alpha_{FF} + 1.1) \cdot f_{FF}^{CO_2} \tag{4}$$

and

$$f_{NEE}^{APO} = (\alpha_{NEE} + 1.1) \cdot f_{NEE}^{CO_2} \tag{5}$$

(both processes are not associated with any $N_2$ flux). To the extent that the terrestrial biospheric stoichiometry is $\alpha_{NEE} = -1.1$ 80 (Severinghaus, 1995), the contribution $f_{NEE}^{APO}$ according to Eq. (5) vanishes. This leaves the fossil-fuel APO flux $f_{FF}^{APO}$ as the only contribution on land, and thus makes it accessible to inverse estimation based on atmospheric APO data over continents.





## 2.2 Atmospheric APO inversion

We use an extension of the APO inversion presented in Rödenbeck et al. (2008). As in any atmospheric inversion calculation (Newsam and Enting, 1988), the surface-atmosphere flux field $f$ is estimated by minimizing the mismatch between the
measured atmospheric tracer abundance and the corresponding abundance simulated by an atmospheric tracer transport model using $f$. The mismatch is gauged by a quadratic cost function. The cost function also contains Bayesian a-priori contributions meant to regularize the inversion, akin to Ridge regression (Hoerl and Kennard, 1970). Here we use the CarboScope inversion software described in Rödenbeck (2005, updated).

As the APO inversion of Rödenbeck et al. (2008) was targeting the sea–air oxygen flux, it had degrees of freedom to adjust
$f_{oc}^{O_2}$ only. All other contributions to the total APO flux ($f_{oc}^{CO_2}$, $f_{oc}^{N_2}$, $f_{FF}^{O_2}$, $f_{FF}^{CO_2}$) had been prescribed. In particular, the FF $CO_2$ emissions had been taken from a bottom-up inventory, and the corresponding FF $O_2$ consumption had been calculated from the $CO_2$ emissions assuming a constant global $\alpha_{FF} = -1.4$ (Keeling, 1988). Here, this APO inversion of Rödenbeck et al. (2008) has been extended by

- using a more realistic (in particular, spatially and temporally explicit) FF stoichiometry $\alpha_{FF}$, calculated via Eq. (2) from
the inventory-based a-priori $f_{FF}^{O_2}$ and $f_{FF}^{CO_2}$ fields;

- adding some degrees of freedom (DoF) to adjust the fossil-fuel emissions in Europe as detailed below;

- adding observational records from European continental stations (Sect. 2.3); and

- for inversions of real data, using a regional transport model with higher resolution over Europe, nested into the global inversion (Appendix A).

The a-priori $f_{FF}^{O_2}$ and $f_{FF}^{CO_2}$ fields are taken from GridFEDv2022.2 (Jones et al., 2022), which uses fuel-type specific oxidative ratios as in Steinbach et al. (2011) to obtain the $O_2$ consumption from the $CO_2$ emissions. The additional DoFs scale the fossil-fuel fluxes on land away from their a-priori values (fossil-fuel emissions over the ocean, being much smaller than those on land, are not scaled). Both $f_{FF}^{O_2}$ and $f_{FF}^{CO_2}$ are scaled by the same factors, such that the FF stoichiometry $\alpha_{FF}$ of the a-posteriori estimate remains identical to that of the prior. The adjustable scaling factors are allowed to vary in space (with an
isotropic correlation length scale of about $380\,\mathrm{km}$) and in time (with a correlation length scale of about a year [filter "Filt1T" in CarboScope notation]; yearly time scale means that the seasonal and any faster variations of the prior remain essentially unchanged). The scaling of the fossil-fuel fluxes is only done within Europe west of $35°\,E$ and north of $33°\,N$ (coloured area in Fig. 1). We verified that the results for the European fossil-fuel fluxes presented in this paper do not appreciably depend on whether or not the inversion set-up also includes any DoFs to scale the non-European fossil-fuel emissions (tests not shown).
APO fluxes in non-European land regions cannot be constrained anyway because almost no continental APO data exist outside Europe so far.

Further technical details of the APO inversion are given in Appendix A.


## 2.3 APO data

The APO inversion has been constrained by data from a set of global stations plus additional stations in central and north-western Europe (Table 1). The locations of the European stations can be seen in Fig. 1 (symbols).

Fig. 2 shows the periods over which APO data are available. The blue vertical dotted lines delimit the analysis period 2007–2021 (inclusive) of this study, chosen to comprise the years when essentially all stations have data, such that temporal variations can be constrained. The influence of gaps in the data records (such as the record at WAO starting mid 2010 or at OXK ending mid 2019) is assessed by additional model runs omitting the individual stations in turn.

Most stations provide flask measurements about once per week. Where hourly in-situ time series are available (here only at WAO), only day-time values (11:00–17:00 local time) have been used, because during this time the atmospheric transport model is expected to have lowest uncertainty due to the well-mixed atmospheric conditions.

At some stations, the data have specifically been selected or adjusted, as described in Appendix B.

## 2.4 Diagnostics

In order to test to what extent the decadal trend in FF $CO_2$ emissions can be estimated by the APO inversion independently of the trend already present in the bottom-up inventory used as a prior, we performed test inversions using a manipulated (counterfactual) prior

$$f_{\text{FF,pri,manip}}^{CO_2}(t) = f_{\text{FF,pri}}^{CO_2}(t) \cdot \begin{cases} 1 + \beta \cdot (t - t_0), & t \geq t_0 \\ 1, & t < t_0 \end{cases} \quad (6)$$

with a scaling factor increasing linearly by $\beta = 1\%/\text{yr}$ starting at $t_0 =$2007-01-01. The increase has been chosen such that the decadal emissions reduction in Western European is partly compensated. The oxygen consumption is scaled in the same way, such that the stoichiometry $\alpha_{\text{FF}}$ of the manipulated prior is identical to that of the original prior. Due to the manipulation, after 10 years (on 2017-01-01, towards the end of our analysis period) the counterfactual prior has emissions $10\%$ higher than the original FF inventory. We will consider to what extent the inversion is able to compensate this scaling on the basis of the information in the APO data.

The manipulated prior has mainly been used in synthetic inversions. These are based on pseudo-data effectively created by a forward transport model simulation from the original a-priori fluxes, to be compared with the original a-priori fluxes as a "known truth". As a complement, we also ran test inversions (based on the real data) using the manipulated prior, to be compared with the results of the main inversion using the original prior.



## 3 Results

### 3.1 FF emissions estimated from available APO data

The dark blue line in Fig. 3 represents the results of our base APO inversion, scaling the $CO_2$ emissions of the bottom-up inventory (black) based on the APO data from station sets A+B (Table 1, symbols in Fig. 1). Similarly, the other color lines show inversion results where the APO data from each individual station in set B were discarded in turn.

On the year-to-year time scale, the inversion estimates contain considerably more variations than reported in the FF inventory. If taken at face value, this would suggest corrections clearly exceeding the expected uncertainty of the inventory. However, the set of runs reveals that most year-to-year features only depend on a single one of the measurement stations. For example, the steeper decreasing trend after about 2016 solely depends on the signals from the WAO station, while the lower fluxes at the beginning of the period and in about 2012 decisively depend on the OXK station. The variability is unlikely to be caused by the incomplete data coverage (Fig. 2) because large anomalies also exist within 2011–2018 when the data records are mostly available. The large variability calls for great care in interpreting the $CO_2$ emissions estimated from the currently available data.

Concerning decadal trends, the large year-to-year variability currently precludes calculating a statistically significant trend. Even if calculating a temporal slope anyway, we find a strong dependence on the chosen stations.

The strong dependence of flux variability on individual stations may reflect local signals spuriously interpreted as regional signals, but also measurement errors, transport model errors, or gaps in the data records. Trying to disentangle these possible influences, we next investigate whether the information content of the data and the degrees of freedom built into the inversion set-up would in principle allow us to estimate decadal trends in the European fossil-fuel emissions.

### 3.2 The potential of available and additional APO measurement stations

Because of the very small network of currently available stations with APO data, we assessed the potential of several station sets using inversions with synthetic data. As synthetic inversions do not involve measurement or model errors, they isolate the effects of data information content and degrees of freedom. The synthetic inversions start from a manipulated (counterfactual) prior described in Sect. 2.4 and try to reconstruct the bottom-up inventory ("known truth") through the information contained in the synthetic data.

According to Fig. 4, the station set A+B can reduce the gap between the counterfactual prior (gray) and the "known truth" (black) in Western Europe by more than a third (solid blue). This is consistent with the behaviour of the real-data inversion if starting from the counterfactual prior as well: The difference in a-posteriori estimates is reduced by about the same factor compared to the difference in prior (tests not shown). In contrast to station set A+B, a synthetic inversion with set A alone does not move the result away from the counterfactual prior (test not shown).





While Fig. 4 considers the average over the Western European region, parts of this area may in fact be better constrained. To investigate this, Fig. 5 shows the local metric

$$\varrho(x,y) = \frac{f_{\mathrm{FF,est}}^{\mathrm{CO_2}}(x,y,t_{\mathrm{x}}) - f_{\mathrm{FF,pri,manip}}^{\mathrm{CO_2}}(x,y,t_{\mathrm{x}})}{f_{\mathrm{FF,true}}^{\mathrm{CO_2}}(x,y,t_{\mathrm{x}}) - f_{\mathrm{FF,pri,manip}}^{\mathrm{CO_2}}(x,y,t_{\mathrm{x}})} \cdot 100\% \tag{7}$$

where $f_{\mathrm{FF,true}}^{\mathrm{CO_2}}$ denotes the "known truth" identical to the normal prior, $f_{\mathrm{FF,pri,manip}}^{\mathrm{CO_2}}$ the manipulated (counterfactual) prior, and $f_{\mathrm{FF,est}}^{\mathrm{CO_2}}$ the synthetic estimate. By construction, this metric equals $100\%$ at pixels where the estimate fully reconstructs the "known truth" despite the deviation of the prior (indicating full constraint), and drops towards zero where the estimate remains close to the prior (indicating absence of data constraint). The metric has been calculated from yearly averaged fluxes at the test time $t_{\mathrm{x}}$ =2017. This test time has been chosen because this is the most recent year with more or less complete data coverage at the European station set B (see Fig. 2). In the base case (station set A+B and standard uncertainty settings), Fig. 5 (top left) reveals that the strongest constraint ($\varrho \approx 50\%$) is found in the area in-between the stations of set B. This is expected because it is the gradients between the stations that determine the fluxes at any location. However, $\varrho$ may also be enhanced there because of the large FF emissions in this same area, which allow the inversion to achieve a particularly large change in fluxes for the same change in scaling factors; indeed we find somewhat larger $\varrho$ values in the same area even if only one of the stations has been used (additional tests not shown).

The synthetic-data inversions also allow testing inversion set-ups with less regularization, such that the FF emissions have more freedom to be adjusted to match the atmospheric data. Such a set-up could not be used with the real data so far, because the increased freedom leads to even larger year-to-year variations than in Fig. 3, being even less likely to be true (not shown). As the synthetic-data inversions are less affected by transport model errors, measurement errors, or local signals, however, they do not have this problem. As shown by the broken blue line in Fig. 4, a synthetic-data inversion with increased freedom is better able to reconstruct the "known truth" from the data: Multiplying the a-priori uncertainty of the DoFs adjusting the FF fluxes by 8 ("8-fold sigma") already matches the "known truth" almost completely in the regional average over Western Europe. This confirms that there is no fundamental obstacle against estimating FF emissions in European subregions from a logistically feasible set of atmospheric stations like set B.

In the spatially explicit view of Fig. 5 (top right), using the more free inversion set-up (8-fold sigma) is seen to achieve essentially perfect reconstruction ($\varrho \approx 100\%$) almost everywhere in-between the stations of set B. Some pixels slightly above 100% compensate others slightly below 100%, as the data can only constrain the sum of fluxes along the path of the air from station to station.

The synthetic-data inversions further allow assessing how the constraint could be improved by additional stations from which APO data are not yet available. We test a set C of European stations where APO measurements have recently been started by the ICOS infrastructure but are not yet available over a period sufficiently long for the inversion. In the synthetic inversion, we assume that data at the stations of set C were already available throughout the inversion calculation (one flask on day 5, 10, 15, 20, 25, and 30 of each month, local noon). The additional stations clearly improve the match in the Western European FF emissions estimates, even for the tighter regularization of the standard set-up (Fig. 4, orange). In the area surrounded by the highest station density, the local constraint reaches 100% (Fig. 5, middle left).





In the less regularized set-up (8-fold sigma), the additional stations of set C slightly further homogenize the match in space (Fig. 5, middle right compared to top right). The additional stations located further away (Scandinavia, Mediterranean) extend

the constrained area, though the gain in power to match the "known truth" seems to stay behind the central area. Possibly, the geometry of the station set C in Northern and Southern Europe does not sufficiently embrace larger emissions hotspots by covering upwind and downwind locations, as it does not have much West-East coverage at any given latitude. However, the smaller gain in $\varrho$ may also just reflect the lower FF emissions (see above). The more freedom is given to the inversion, the better the performance is also in these areas (not shown). Adding further stations outside the geographical range of set C

(station set D consisting of ICOS class 2 stations where APO measurements are not yet planned, Fig. 5, bottom right) similarly extends the better-constrained area towards Wales and the Iberian Peninsula.

  As a general pattern in the various synthetic inversion results, the addition of stations and the decrease in regularization strength have roughly similar effects. This is because additional data also exert a greater total power in the cost function. From our present results, therefore, we cannot yet make firm statements on a desirable station density. Such statements would also

depend on the actual heterogeneity of the FF emissions signals, which in turn would need to be reflected properly in the chosen a-priori spatial correlation lengths in the inversion calculation (here we only picked a correlation length rather arbitrarily). A deeper investigation seems only possible in a few years from now when multi-year time series observed at the stations of set C will be available.

  As the ocean fluxes in the "known truth" have been chosen identical to the ocean prior (Sect. 2.4), any deviation of the

a-posteriori ocean fluxes away from the prior are spurious signals misattributed from land where prior and "known truth" are different. Among our synthetic-data inversion runs with different station sets and uncertainty settings, those that reconstruct the FF fluxes on land less well also deviate more in their ocean APO fluxes (not shown). This illustrates the presence of a-posteriori anti-correlations between land and ocean fluxes, limiting the capability to correctly attribute the atmospheric APO signals to land or ocean. This means that part of the variability in Fig. 3 may also be misattributed ocean variability. The situation is

consistent with forward transport simulations by Chawner et al. (2023) who find the influence of oceanic APO signals at UK stations including WAO to partially be as strong as the signals related to FF emissions. An incomplete capability to separate land and ocean fluxes is also a widely known limitation of atmospheric inversions in general (Peylin et al., 2013).

### 3.3 Uncertainties related to the $O_2:CO_2$ stoichiometry of FF emissions

The APO data primarily constrain the FF-related APO flux $f_{\mathrm{FF,est}}^{\mathrm{APO}}$. In our inversion algorithm, this APO flux is linked in a fixed

way to the FF $CO_2$ emissions $f_{\mathrm{FF,est}}^{\mathrm{CO_2}}$ via Eq. (4) using the $O_2:CO_2$ stoichiometry $\alpha_{\mathrm{FF,pri}}$ as given in the bottom-up inventory. We can however re-compute the FF $CO_2$ emissions from the estimated APO flux $f_{\mathrm{FF,est}}^{\mathrm{APO}}$ under the assumption of modified stoichiometries $\alpha_{\mathrm{FF,mod}}$ by

$$f_{\mathrm{FF,mod}}^{\mathrm{CO_2}} = \frac{1}{\alpha_{\mathrm{FF,mod}} + 1.1} \cdot f_{\mathrm{FF,est}}^{\mathrm{APO}} \tag{8}$$

In the uncertainty assessment of this section, we used the inventory as a proxy for an APO inversion result, as the actual

inversion results of Fig. 3 seem to noisy for this purpose.



Fig. 6 compares the FF $CO_2$ emissions $f_{\text{FF,est}}^{CO_2}$ as it would directly come from the APO inversion using the original $O_2$:$CO_2$ stoichiometry $\alpha_{\text{FF,pri}}$ (solid blue) with recomputed FF $CO_2$ emissions $f_{\text{FF,mod}}^{CO_2}$ assuming constant stoichiometries $\alpha_{\text{FF,mod}}$ of $-1.50$ (green) or $-1.55$ (cyan), respectively. We first notice that the solid blue line roughly agrees with the green one in the earlier years but approaches and even crosses the cyan line in the later years. This reflects that the FF stoichiometry $\alpha_{\text{FF,pri}}$

according to the bottom-up inventory (in a flux-weighted average sense for "Western Europe") changed from about $-1.50$ to beyond $-1.55$ over the course of our analysis period, compatible with the increasing share of natural gas in the fuel mix. It also shows, however, that a shift in the assumed FF stoichiometry $\alpha_{\text{FF}}$ by the same $0.05$ difference would alter the inferred FF $CO_2$ emissions by an amount more than half of the emissions reduction over this period. As noted above, this alteration in the $CO_2$ emissions estimate occurs despite the estimated APO flux $f_{\text{FF,est}}^{\text{APO}}$ being identical for all three lines of Fig. 6. This means

that the FF stoichiometry $\alpha_{\text{FF,pri}}$ and its temporal changes need to be known to an uncertainty well below $0.05$.

### 3.4 Uncertainties related to the $O_2$:$CO_2$ stoichiometry of exchanges from the land biosphere

According to Eq. (5), the APO flux $f_{\text{NEE}}^{\text{APO}}$ associated with photosynthesis and respiration on land vanishes *only* if the stoichiometry is exactly $\alpha_{\text{NEE}} = -1.1$ as assumed in the APO definition (Stephens et al., 1998). However, recent measurements of atmospheric $O_2$ and $CO_2$ variations closer to or within the canopy suggest that the actual stoichiometry $\alpha_{\text{NEE}}$ may be smaller in

absolute value and depend on time of day and vegetation state (e.g., Seibt et al., 2004), even though it remains unclear what this means for the large-scale effective stoichiometry relevant here. As the APO data constrain the sum of all surface–atmosphere APO fluxes, any presence of a non-zero $f_{\text{NEE}}^{\text{APO}}$ would lead to a compensating bias $\Delta f_{\text{FF,est}}^{\text{APO}} \approx -f_{\text{NEE}}^{\text{APO}}$, which translates into a bias

$$\Delta f_{\text{FF,est}}^{CO_2} \approx -\frac{\alpha_{\text{NEE}} + 1.1}{\alpha_{\text{FF}} + 1.1} \cdot \text{NEE} \tag{9}$$

in the estimated FF $CO_2$ emissions. We calculated this bias for an assumed biospheric stoichiometry $\alpha_{\text{NEE}} = -1.05$, using $\alpha_{\text{FF}} = -1.55$ (the approximate current value for "Western Europe", see Sect. 3.3) and an NEE flux field estimated from atmospheric $CO_2$ data (CarboScope atmospheric $CO_2$ inversion `s10oc_v2022`, update of Rödenbeck et al., 2018).

Again using the inventory as a proxy for an APO inversion result, Fig. 7 compares the estimated FF $CO_2$ emissions without (blue) and with (green) this bias added. On the yearly time-scale (top) mainly considered in this study, the bias only leads

to a small shift compared to the year-to-year variations or decadal trend. Therefore, uncertainties in the biospheric $O_2$:$CO_2$ stoichiometry on the order of $0.05$ should not be problematic in the APO-based estimation of yearly FF $CO_2$ emissions and their trend.

For seasonal variations however (Fig. 7 bottom), due to the large seasonal cycle of NEE, the impact of the associated bias turns out to be as large as the signal itself. To the extent that the chosen shift in the biospheric $O_2$:$CO_2$ stoichiometry by

$0.05$ represents its uncertainty, this implies that our APO-based estimates cannot make any meaningful statements on the seasonality of FF $CO_2$ emissions. (If the real $O_2$:$CO_2$ stoichiometry of NEE would significantly change with seasons, there might additionally be some rectification of the seasonal NEE bias, which could even have an effect on the yearly mean bias. We do not know so far whether this is the case.)





## 4 Discussion

Many of the results presented here are encouraging regarding the use of APO data to constrain yearly FF $CO_2$ emissions
in European subregions. Our inversions with synthetic data demonstrate that a logistically realistic set of stations suffices to
constrain large-scale emissions including decadal trends (Sect. 3.2). Though the actual $O_2$:$CO_2$ stoichiometry $\alpha_{NEE}$ of land-
biospheric fluxes may deviate from the value of 1.1 assumed in the APO definition (see Eq. (1)) and rather be in the range
of 1.0 to 1.1 and possibly beyond (see a summary of values in Keeling and Manning, 2014), this does not seem to pose a

fundamental obstacle on the yearly time scale (Sect. 3.4); nevertheless, a better characterization of $\alpha_{NEE}$ on the space and time
scales relevant here would be useful for accurate FF emission estimation from APO data.

More problematically, uncertainties in the $O_2$:$CO_2$ stoichiometry $\alpha_{FF}$ of the FF emissions need to be well below 0.05 on
the regional scale (Sect. 3.3). Keeling and Manning (2014) (their Tab. 2) give uncertainties of the $O_2$:$CO_2$ stoichiometry for
individual fuel types of $\pm 0.03$ or $\pm 0.04$. In the $\alpha_{FF}$ values used here, calculated from the $CO_2$ and $O_2$ fluxes from the bottom-

up inventory, there is additional uncertainty related to possible errors in the fuel mix. Thus, more work is needed to determine
the $O_2$:$CO_2$ stoichiometry of FF emissions sufficiently accurately.

The largest problem to date are strong interannual variations implied by the APO data available within Europe for the past
1–2 decades (Sect. 3.1). Even in our standard APO inversion set-up where adjustments of the FF emissions are regularized so
strongly that –according to our synthetic inversion– it becomes difficult to constrain trends, the implied interannual variations

still exceed the expected uncertainty of the bottom-up inventory. We therefore consider these strong interannual variations as
spurious. Unfortunately, they mostly mask the information on decadal trends.

As in any atmospheric inversion, spurious interannual variations could conceivably arise from systematic time-dependent
measurement or transport model errors, or from small-scale signals being interpreted by the inversion as regional signals. If
systematic measurement errors or local signals played a role, we would expect them to depend on the use of specific stations.

Indeed, we did not find much coherence between the variations inferred from the small number of existing European stations
(Sect. 3.1). Unfortunately, despite our attempts for example looking at the model-data residuals of the inversion runs using
different station sets, we did not find any specific clues how to tackle this problem.

We expect that clearer answers can be given as soon as the upcoming or recently started APO data from the ICOS stations
(set C) will cover several years. To the extent that the variations reflect random errors in data and transport model, the larger

station set will help to average these out. Possibly, the availability of more stations will also allow us to establish the reliability
of the variations by confirming that they are resolved using different independent combinations of stations (as in Rödenbeck
et al., 2008). This approach requires redundancy in the network, i.e. the ability of different combinations of stations to resolve
fluxes over essentially the same domain. Unfortunately, the coverage at present is too sparse.

A larger set of APO measurement stations will also more adequately represent the spatial heterogeneity of FF emissions and

their trends. In our tests based on a counterfactual prior, we only implemented a spatially uniform deviation from the original
inventory. In reality, the decadal trends of FF emissions are spatially very inhomogeneous (not shown) reflecting the build-up or
decommissioning of industrial infrastructure in the course of time. Consequently, the errors of bottom-up inventories may also



be expected to be spatially heterogeneous, as the proxy data used to disaggregate country totals will not perfectly reflect such changes. By neglecting this heterogeneity, our tests pose an easier challenge to the inversion. The issue should be revisited in

the light of the upcoming more dense station network in a few years from now. Such work should also include tests to find out which spatial and temporal correlations are appropriate to be implemented in the inversion set-up, as a denser station network may allow to constrain degrees of freedom on a less coarse spatial resolution. We note that this situation also applies in general to other top-down methods for FF quantification, such as radiocarbon. This means that development work regarding any of these tracers may have co-benefits for other tracers as well.

By using a flux representation on daily time steps, our calculation neglects possible complications related to diurnal variations. Diurnal co-variations in fluxes and atmospheric transport may lead to biases also on longer temporal scales (often called "rectification effects"). Similarly, biases may arise from diurnal co-variations in fluxes and the oxidative ratios ($\alpha_{FF}$ or $\alpha_{NEE}$). Unfortunately, there is no easy solution, because the diurnal variations in most fluxes and oxidative ratios can neither be inferred from the atmospheric data nor are they well known a-priori. More work is needed to quantify and possibly correct the

impact of the diurnal rectification effects on the FF emissions estimates. However, regarding decadal trends as targeted in this paper, these effects may be less problematic to the extent that the diurnal variations stay similar over time.

A note on $CO_2$ emissions from cement production: These emissions are somewhat special among the anthropogenic emissions as there is no associated $O_2$ consumption. However, as they are part of the bottom-up inventory, they are properly reflected in the overall stoichiometry $\alpha_{FF}$ used here. In Europe, the share of cement production in the total anthropogenic $CO_2$ emissions

is low anyway; however this issue may need more attention for APO-based studies in other parts of the world.

A possible complication in future uses of continental APO data may be the production of hydrogen ($H_2$) as a fuel, as the electrolysis of water also produces $O_2$. Though the same amount of $O_2$ is consumed later in $H_2$ oxidation during fuel use, its location will generally differ from that of $H_2$ production and thus have an impact on the APO observations.

## 5 Conclusions

From the assessments presented here we conclude that the estimation of decadal trends in fossil-fuel $CO_2$ emissions in subcontinental regions from sustained APO measurements on at least weekly frequency should be feasible in practice. Even though our estimates based on a small number of existing measurement locations involved too large apparent year-to-year variations to infer decadal trends, more stable estimates seem possible as soon as a denser network of APO observations over several years will be available. We therefore believe that ongoing measurement efforts, including those recently started within the European

ICOS research infrastructure, are valuable investments in future capabilities to independently verify fossil-fuel $CO_2$ emissions inventories.



## Appendix A: Technical details of the inversion

Although this study is only considering fluxes within Europe, the inversion of atmospheric data is an intrinsically global problem. Due to the computational requirements, global inversions as in Rödenbeck et al. (2008) are limited to relatively coarse resolution of the atmospheric tracer transport simulation, involving relatively large model errors. Regional inversions offer a way to increase the resolution within the target region, at the cost of additional complexity to properly transfer the global signals across the regional boundary. Further, if simulating the regional atmospheric transport via pre-computed "footprints" as done here (see Appendix A2), additional hypothetical data points as in our synthetic inversions cannot easily be included. Therefore, all synthetic inversions (Figs. 4 and 5, being not very sensitive to transport errors anyway) have been done as global calculations in this study, while the results using real APO data (Fig. 3) have been refined by regional nesting.

## A1 Global inversion

The global inversion has been done with the TM3 atmospheric transport model (Heimann and Körner, 2003) on a resolution of $5°$longitude $\times$ about $4°$latitude. TM3 has been driven by meteorological fields from the NCEP reanalysis (Kalnay et al., 1996). The pixel size of the global flux field is $2.5°$longitude $\times 2°$latitude.

In the previous set-up (version `v2021`) of the APO inversion, the a-priori uncertainties of the long-term and seasonal DoFs of the oceanic APO flux had been enhanced compared to the non-seasonal interannual DoFs. This uncertainty enhancement has been removed, as it had worsened the mutual misattribution of land and ocean signals as revealed by synthetic-data tests (not shown). Using non-enhanced uncertainties for the oceanic seasonal cycle DoFs also improves the agreement with the extra-tropical seasonal cycles of the independent ocean $O_2$ fluxes by Garcia and Keeling (2001) based on heat fluxes (not shown).

As in Rödenbeck et al. (2008) (and in the CarboScope inversions in general, Rödenbeck, 2005), the Bayesian uncertainty of the model–data mismatch has been chosen in dependence of an assumed model uncertainty class of each station (Table 1) as R (remote): $1.5\,\mathrm{ppm}$, S (shore): $2.25\,\mathrm{ppm}$, M (mountain): $2.25\,\mathrm{ppm}$, T (tower): $4.5\,\mathrm{ppm}$, C (continental): $4.5\,\mathrm{ppm}$, added quadratically to a sweepingly assumed measurement uncertainty of $0.4\,\mathrm{ppm}$. This value, slightly larger than in the CarboScope $CO_2$ inversion, may seem small for APO measurements, but has been kept nevertheless in light of the model uncertainty being larger and barely known anyway. Further, a "data-density weighting" (Sect. 2.3.3 of Rödenbeck, 2005) has been applied, assuming that data points within time intervals of a week may not add new information due to the temporal correlations in synoptic transport and in the flux representation. This data-density weighting also leads to comparable weights of flask or in-situ stations despite their dramatically different data density.

Finally, data outliers have been removed by an "$n\sigma$ selection" (similar to an "Iteratively Re-weighted Least Squares" algorithm). As described in Sect. A1.3 of Rödenbeck et al. (2018), pre-runs of the inversion based on all available data points have been done. Separately for every station, the standard deviations ($\sigma$) of the a-posteriori residuals of these pre-runs have been determined, and any data point with a residual larger than $n\sigma$ has been discarded in all the main inversion runs. In contrast to Rödenbeck et al. (2018), a slightly stricter threshold of $n = 1.5$ (rather than $n = 2$) has been used. Moreover, the selection of





the individual stations of set B has been done according to separate pre-runs using station set A and only the considered station
of set B. This was done in light of the partially contradicting signals from the individual stations (see Fig. 3) which would
otherwise also remove data points that cannot be fit because the inversion has to compromise between the stations.

## A2 Regional inversion

In order to reduce transport model uncertainty at the continental stations of set B, we also performed higher-resolution regional
inversions within our target region, using the regional transport model STILT (Lin et al., 2003; Trusilova et al., 2010). The
regional flux field covers $15°$ W–$35°$ Eand $33°$ N–$73°$ Nwith a pixel size of $0.25°$ longitude $\times$ $0.25°$ latitude.

The regional inversion has been nested into the global inversion using the "two-step scheme" introduced in Rödenbeck et al.
(2009). Each regional inversion uses the far-field contributions from a global inversion run based on the same set of stations.
(In principle, one may consider to use the same far-field contributions [e.g., from a global inversion based on the maximum
station set] for all regional runs. This does change the result, even though by a difference smaller than the difference between
the station sets in Fig. 3 [tests not shown]. We chose identical station sets in global and regional inversion runs as having the
greatest mutual consistency.)

In terms of data, a-priori fluxes, and uncertainties, the regional inversion is done like the global inversion. We also take over
the result of the "$n\sigma$" outlier detection from the global runs, even though there is a chance that some data points discarded for
not being well represented by TM3 can now be better represented by STILT and thus could be kept. However, using far-field
contributions simulated by TM3 for these data points may also pose a risk. Unfortunately, the choice of outlier detection in the
regional inversion does appreciably influence its results (tests not shown), comparable to the choice of station set in Fig. 3.

## Appendix B: Data selection and adjustments

### B1 Data by UEA

At WAO station (Adcock et al., 2023), all values flagged 2 (insecure values during build-up phase until May 2010) or 3
(contamination due to a leaking pump from 2018-03-01 to 2019-03-22) have been discarded.

### B2 Data by BGC

In order to ensure data quality and comparability, the BGC-IsoLab measurements (see Table 1) have been compared to those
done from regular simultaneous sampling by the Scripps laboratory (SIO) at the supersite Alert (ALT). SIO is recognised as
the expert laboratory for $O_2/N_2$ measurements within the Global Atmospheric Watch community. The comparison revealed
two issues that have been addressed in the following ways:

(1) The "Ar-corrected" $O_2/N_2$ data as described in Heimann et al. (2022) have been used. This adjustment attempts to remove
influences from leakages during flask storage or from other processes that affect $O_2/N_2$ and $Ar/N_2$ in proportional ways.
Indeed, the "Ar-corrected" $O_2/N_2$ data agree better than the uncorrected $O_2/N_2$ data with the simultaneous measurements





by SIO at station ALT (Fig. 10 (middle panel) of Heimann et al., 2022). Of course, the Ar-based adjustment also leads to a spurious transfer of real signals from the $Ar/N_2$ values to the $O_2/N_2$ values; however, the benefit of reducing measurement artifacts seems to outweigh this problem.

(2) Even after the "Ar-correction" has been applied, the BGC $O_2/N_2$ values still differ from the SIO values in a systematic way. We find both an overall mean difference, and a rectangular-shaped enhancement of the difference during about 2015–2019

(Fig. 8). We attribute the overall mean difference to an offset between the Scripps scale and its local implementation through the standard gases at BGC. This offset is currently being determined by re-measuring BGC standard gases in the SIO laboratory, allowing to reduce it in future BGC data releases. As this is an ongoing process, we use for now the SIO-BGC comparison at ALT to apply an additive correction to the BGC APO values for this paper, as described below.

The end of the period of enhanced SIO-BGC differences could be traced to the replacement of a broken autosampler valco

valve in the BGC-IsoLab. Strikingly, all the BGC flasks after the anomalous period have been measured after this replacement on 2020-06-08 (see the color coding of the difference values in Fig. 8 according to the BGC measurement date; note the delay between sampling and measurement due to the remoteness of station ALT). We therefore hypothesise that the systematically larger SIO-BGC differences were caused by a micro-leak of the valco valve that had remained unrecognised. Such a leak can adversely affect the $O_2/N_2$ ratios. As sample and standard gases pass the same valve, one would actually expect that the effect

would cancel out through the calibration. Conceivably, however, the leak affected standards and samples in different ways, as standards are supplied by high pressure tanks, while samples come in 1-liter glass flasks. While the pressure and flowrate of the standards are kept constant, the low sample pressure (between 1.4 and 2.0 bar absolute pressure) of the sampling flasks cause a small pressure gradient, and thus flow gradient, through the valco valve during the measurement process, which requires 200 ml of sample. It is conceivable that a micro leak in the valco valve affected the sample flow subject to a pressure gradient, but

not the standard gas flow that was not subject to a pressure gradient.

The assumption that the entire anomalous period was related to the same cause, is supported by the behaviour of the individual flasks having been sampled at its beginning in late 2014 or early 2015. At each sampling time, generally three flasks are sampled in the BGC program. Coincidentally, individual flasks of these triplets sampled simultaneously have been measured at times several weeks or even months apart. It turns out that flasks measured until early April 2015 tend to agree with the

simultaneously sampled SIO flasks systematically better than flasks measured from early May 2015 onwards. In Fig. 8, the corresponding triplets are recognized by having two difference values for one sampling time, a lower one (green) from the flask(s) measured before 2015-04-20 and a larger one (violet) from the flask(s) measured on or after that day. Strikingly, the lower/higher difference values have a similar order of magnitude as those outside/inside the anomalous period. We therefore consider 2015-04-20 as the start date of the "leakage period" (even though the available information would also be compatible

with the leak having occurred about 1 week earlier or later). The last measurements before the replacement of the valve were done on 2020-06-04, thus marking the end of the "leakage period".

If the step changes in the SIO-BGC difference at station ALT were caused by a leaking valve rather than a specific sampling problem at this station, the data from the other BGC stations must be affected in the same way. Indeed, test inversions using



set A plus the individual BGC stations (before the data correction described below has been applied) yield a similar downward
drop in estimated APO fluxes within the area of influence of the respective stations (not shown).

To compensate for the effect of the leak, we split the entire set of individual flask values (for all BGC stations) into two
distinct sets having been measured within or outside the "leakage period" 2015-04-20 through 2020-06-04, respectively. For
these two sets, we separately calculate the mean SIO-minus-BGC difference at ALT (discarding any difference values with
sampling times before 2008), and add it as a correction to all triplet means from the corresponding set (outside the "leakage
period": 1.30 ppm, inside: 2.89 ppm). Finally, we re-unify the two separate time series of each station (note that for all stations
except ALT, every flask triplet has been measured either completely within or completely outside the "leakage period", such
that we do not have to deal with handling more than one triplet mean at any given sampling time for the stations used in the
inversion).

## B3  Data by NIES

In order to bridge the differences in scale between NIES and SIO, the data by NIES (Tohjima et al., 2019) have been adjusted
according to

$$\delta(O_2/N_2)_{SIO} = 1.024 \cdot \delta(O_2/N_2)_{NIES} \tag{B1}$$

$$+ \{-0.28\,\mathrm{permeg\,yr^{-1}} \cdot (t - t_0) - 197.1\,\mathrm{permeg}\}$$

The factor in the first line represents span differences, taken from the scale comparison by Aoki et al. (2021) based on measurements against a gravimetric scale. The time-dependent function in the second line represents scale drifts, obtained by linear fit
to the differences between regular simultaneous measurements by NIES and SIO at La Jolla (California) from $t_0 =$ 2010-03-02
through 2022-05-15.

*Data availability.*  Inversion results will be made available at https://www.bgc-jena.mpg.de/CarboScope/?ID=apo

*Author contributions.*  CR, SH, and IL conceptualized the study. KEA, ME, AJ, RFK, ACM, HM, PAP, MR, and YT were involved in the
collection, analysis, and curation of the APO data from the various monitoring stations, which form the basis of this work. SM provided
the "footprint" data of the STILT model. CR developed the inversion software, carried out the inversion runs, and visualized the results. All
co-authors, in particular PAP and IL, took part in the analysis and interpretation of the results. CR prepared the manuscript, with important
contributions from all co-authors.

*Competing interests.*  Some authors are members of the editorial board of journal ACP. The peer-review process was guided by an independent editor, and the authors have also no other competing interests to declare.



*Acknowledgements.* We would like to thank all persons involved in the APO measurements. Matt Jones kindly added a $O_2$ layer to the GridFED emissions inventory. Atmospheric $O_2$ and $CO_2$ measurements at WAO were funded by the U.K. Natural Environment Research Council (NERC) grants NE/F005733/1, NE/I013342/1, NE/I02934X/1, QUEST010005, NE/N016238/1, NE/S004521/1, and NE/R011532/1, by the EU FP6 Integrated Project CarboOcean (grant agreement no. 511176 GOCE), and have also been supported by the U.K. National Centre for Atmospheric Science (NCAS) from December 2013 onward. A. Manning, P. Pickers, and K. Adcock also received funding from the European Union's Horizon Europe Research and Innovation programme under HORIZON-CL5-2022-D1-02 Grant Agreement No 101081430 - PARIS.

The service charges for this open access publication have been covered by the Max Planck Society.



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





**Table 1.** Monitoring stations measuring atmospheric $O_2$ and $CO_2$ abundance used in the APO inversions presented here. Stations have been grouped into sets A, B, C, and D, and ordered within each set by either latitude or longitude.

| Station Code | Name, geographic region | Latitude (°) | Longitude (°) | Height (m a.s.l.) | Institution | Type | Mod. Unct. Class [a] |
|---|---|---|---|---|---|---|---|
| *A: Global background stations* | | | | | | | |
| ALT | Alert, Canada | 82.45 | -62.52 | 210 | SIO | flask | S |
| CBA | Cold Bay, Alaska | 55.20 | -162.72 | 25 | SIO | flask | S |
| COI | Cape Ochiishi, Japan | 43.17 | 145.50 | 45 | NIES | flask | S |
| LJO | La Jolla, California | 32.87 | -117.25 | 15 | SIO | flask | S |
| HAT | Hateruma Island, Japan | 24.05 | 123.81 | 10 | NIES | flask | R |
| MLO | Mauna Loa, Hawaii | 19.53 | -155.58 | 3397 | SIO | flask | R |
| KUM | Cape Kumukahi, Hawaii | 19.52 | -154.82 | 40 | SIO | flask | R |
| CVR | Cabo Verde, Atlantic | 16.86 | -24.87 | 10 | BGC | flask | S |
| SMO | Tutuila, Am. Samoa, Pacific | -14.25 | -170.57 | 42 | SIO | flask | R |
| CGO | Cape Grim, Tasmania | -40.68 | 144.68 | 94 | SIO | flask | S |
| PSA | Palmer Station, Antarctica | -64.92 | -64.00 | 10 | SIO | flask | R |
| SPO | South Pole | -89.98 | -24.80 | 2810 | SIO | flask | R |
| *B: European stations (APO data available during analysis period)* | | | | | | | |
| SIS | Shetland Islands, UK | 60.28 | -1.28 | 30 | BGC | flask | R |
| WAO | Weybourne, UK | 52.95 | 1.12 | 10[b] | UEA | in-situ | S |
| JFJ | Jungfraujoch, Switzerland | 46.55 | 7.98 | 3580 | BGC | flask | M |
| OXK | Ochsenkopf, Germany | 50.02 | 11.80 | 1183 | BGC | flask | M |
| BIK | Białystok, Poland | 53.22 | 23.03 | 300[b] | BGC | flask | T |
| *C: European stations (APO data planned or recently started by ICOS)* | | | | | | | |
| ZEP | Zeppelin, Spitsbergen | 78.91 | 11.89 | 474 | synth.[c] | flask | R |
| PAL | Pallas, Finland | 67.97 | 24.12 | 565 | synth.[c] | flask | C |
| SVB | Svartberget, Sweden | 64.26 | 19.77 | 150[b] | synth.[c] | flask | T |
| SMR | Hyytiälä, Finland | 61.85 | 24.29 | 125[b] | synth.[c] | flask | T |
| NOR | Norunda, Sweden | 60.09 | 17.48 | 100[b] | synth.[c] | flask | T |
| HTM | Hyltemossa, Sweden | 56.10 | 13.42 | 150[b] | synth.[c] | flask | T |
| GAT | Gartow, Germany | 53.07 | 11.44 | 341[b] | synth.[c] | flask | T |
| STE | Steinkimmen, Germany | 53.04 | 8.46 | 252[b] | synth.[c] | flask | T |
| LIN | Lindenberg, Germany | 52.17 | 14.12 | 98[b] | synth.[c] | flask | T |
| CBW | Cabauw, The Netherlands | 51.97 | 4.93 | 207[b] | synth.[c] | flask | T |
| KRE | Křešín u Pacova, Czech Republic | 49.57 | 15.08 | 250[b] | synth.[c] | flask | T |
| KIT | Karlsruhe, Germany | 49.09 | 8.42 | 200[b] | synth.[c] | flask | T |
| SAC | Saclay, France | 48.72 | 2.14 | 100[b] | synth.[c] | flask | T |
| OPE | Houdelaincourt, France | 48.56 | 5.50 | 120[b] | synth.[c] | flask | T |
| HPB | Hohenpeissenberg, Germany | 47.80 | 11.02 | 131[b] | synth.[c] | flask | T |
| LMP | Lampedusa, Mediterranean | 35.52 | 12.63 | 45 | synth.[c] | flask | S |
| *D: European stations (ICOS class 2 stations, APO data not planned so far)* | | | | | | | |
| MHD | Mace Head, Ireland | 53.33 | -9.90 | 25 | synth.[c] | flask | S |
| RGL | Ridge Hill, UK | 52.00 | -2.54 | 90[b] | synth.[c] | flask | T |
| HUN | Hegyhátsál, Hungary | 46.95 | 16.65 | 115[b] | synth.[c] | flask | T |
| ARN | El Arenosillo, Spain | 37.10 | -6.73 | 100[b] | synth.[c] | flask | T |
| IZO | Izaña, Canary Islands | 28.31 | -16.50 | 2403 | synth.[c] | flask | R |

[a] Model uncertainty class (C=continental, M=mountain, R=remote, S=shore, T=tower, see Appendix A); [b] Height above ground; [c] Using synthetic data only;

BGC: Max Planck Institute for Biogeochemistry, NIES: National Institute for Environmental Studies (Tohjima et al., 2019), SIO: Scripps Institution of Oceanography, UEA: University of East Anglia (Adcock et al., 2023)



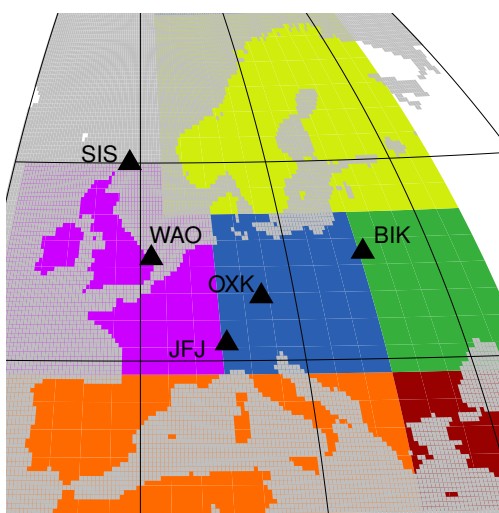

**Figure 1.** Symbols: Locations of the European APO measurement stations used in the APO inversion (set B, see Table 1). Colors: Regions over which the estimated gridded FF fluxes are integrated for time series figures ("Western Europe": magenta).

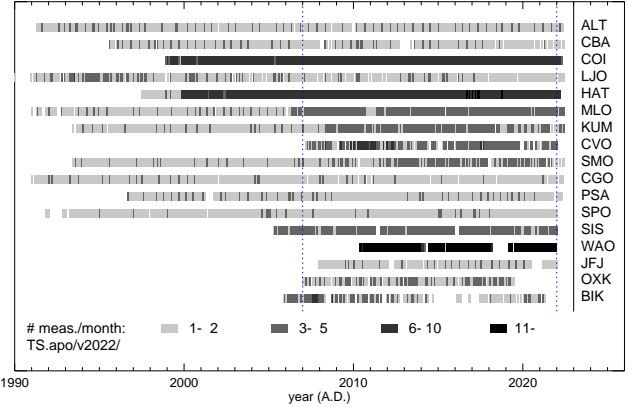

**Figure 2.** Time periods over which APO data are available at the individual stations of sets A and B (Table 1). The vertical dotted lines enclose the period shown in the flux time series figures.





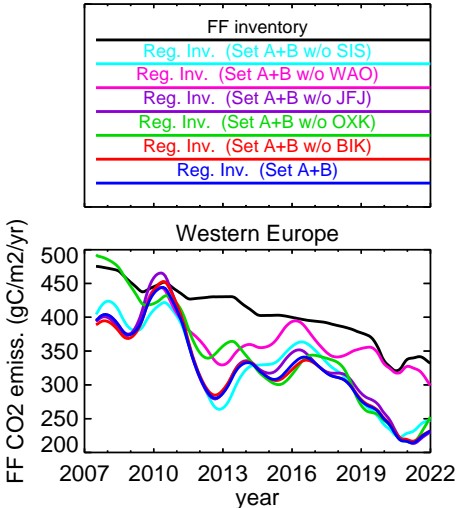

**Figure 3.** Estimated interannual variations in the FF emissions (deseasonalized by running annual averaging and further smoothed on a 3-monthly scale) integrated over the "Western Europe" region of Fig. 1. Dark blue shows the results of the regional APO inversion based on station set A+B, while the other color lines omit individual stations of set B (see Table 1). Black shows the original FF inventory used as a prior.



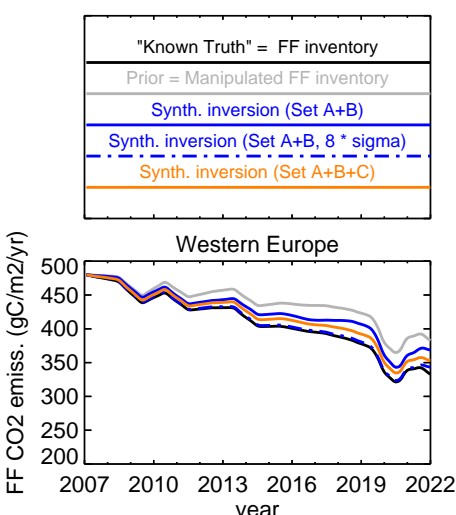

**Figure 4.** Synthetic-data test: Starting from a counterfactual prior with less emissions reduction (manipulated FF inventory defined in Sect. 2.4, grey) the synthetic inversions try to reconstruct the "known truth" (original FF inventory, black) from pseudo-data created by an atmospheric transport simulation for the same locations and times as the real data. The color lines represent synthetic APO inversions using the same inversion set-up as Fig. 3 (solid) or a less regularized set-up (8-fold a-priori uncertainty of the fossil-fuel related degrees of freedom) having more freedom to match the data ("8-fold sigma", broken). The color hue indicates the station set used.



**Figure 5.** Maps of the agreement $\varrho$ of the synthetic inversions as in Fig. 4 with the "known truth", relative to the difference between the "known truth" and the prior (Eq. (7)). Values around 100% (gray) indicate complete correction of the counterfactual prior. The black symbols indicate the European APO measurement stations used in the respective run



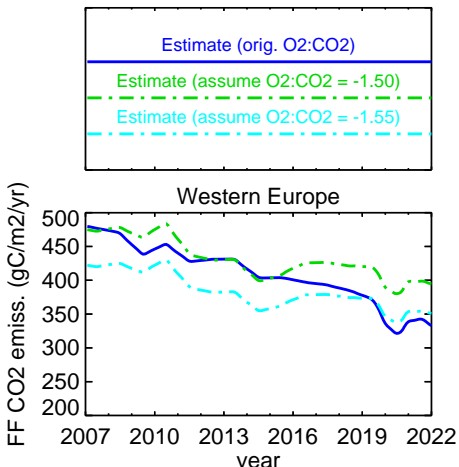

**Figure 6.** Importance of the $O_2$:$CO_2$ stoichiometry of FF emissions ($\alpha_{FF,pri}$) used in the APO inversion: A given FF $CO_2$ emissions estimate based on the original time and space varying $\alpha_{FF,pri}$ from the bottom-up inventory (blue, here using the FF inventory as a proxy for an inversion result) is compared to $CO_2$ emissions re-computed from the corresponding APO flux assuming constant stoichiometries of $-1.50$ (green) or $-1.55$ (cyan), respectively.

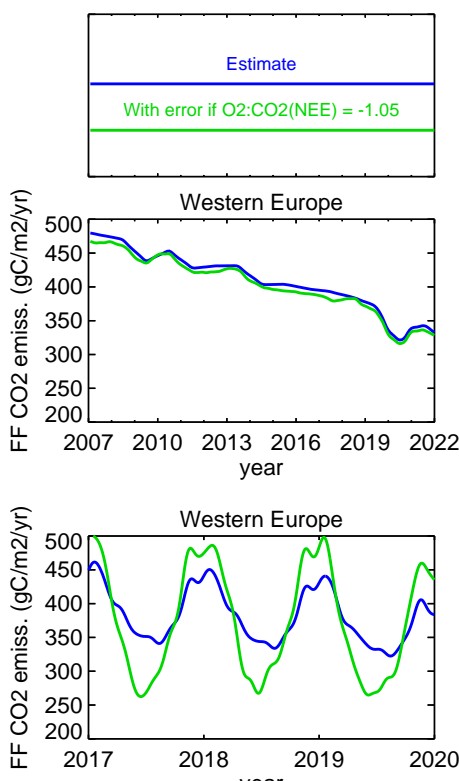

**Figure 7.** What if the true $O_2$:$CO_2$ stoichiometry of terrestrial biosphere was $-1.05$, rather than $-1.1$ as assumed in the APO definition? For a given FF $CO_2$ emissions estimate (blue, here using the FF inventory as a proxy for an inversion result) the difference between the green and blue lines represents the corresponding error as calculated from Eq. (9). Top: yearly fluxes as in Fig. 3; bottom: fluxes on the original daily time resolution for 3 example years.

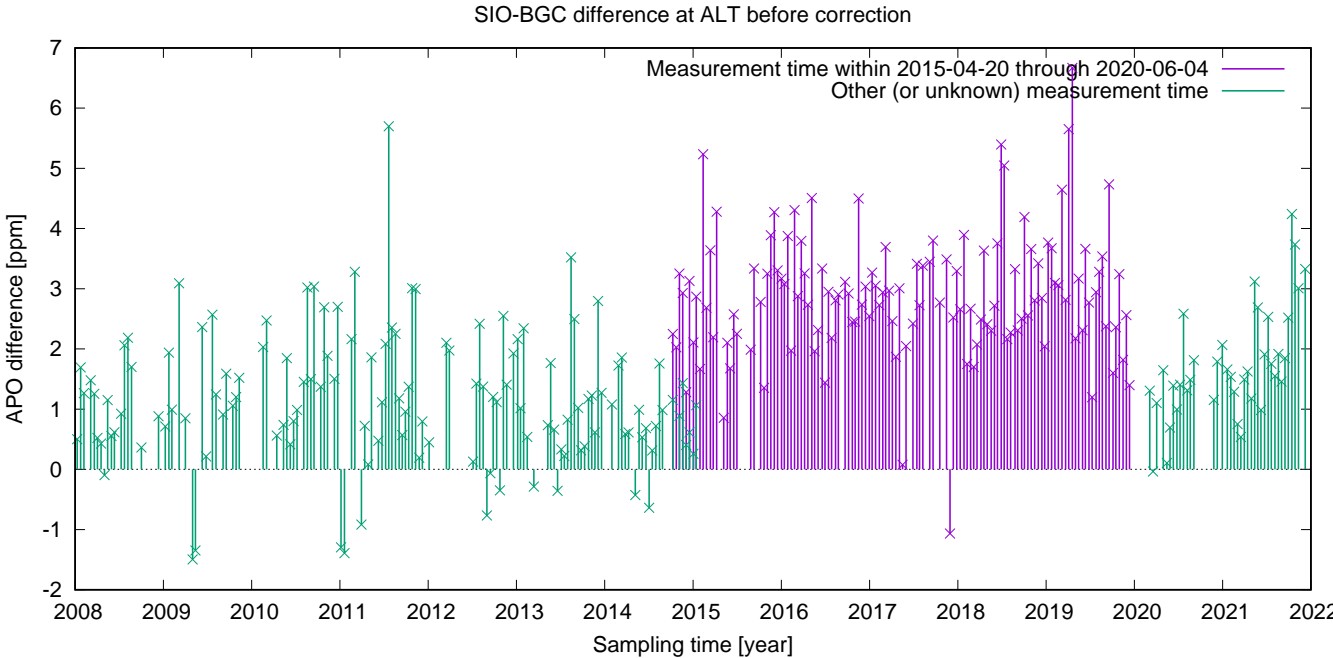

**Figure 8.** Difference between the APO observations (flask triplet means, before additive correction) sampled simultaneously by SIO and BGC at ALT station. The triplet means of the BGC flasks have been calculated separately for flasks with measurement times within or outside the "leakage period" 2015-04-20 through 2020-06-04(see Sect. B2 for more explanation and the corresponding correction applied).