# Peer review of "The suitability of atmospheric oxygen measurements to constrain Western European fossil-fuel $CO_2$ emissions and their trends"

_EGUsphere, 2023_

## Author Response (AR1)

**Reply to Reviewer comment RC1**

We thank the reviewer for their helpful comments on our manuscript. In the following, the original reviewer comments are given in italics.

*An inverse modeling study of European fossil fuel CO2 emissions is presented using oxygen and carbon dioxide measurements. The study uses Atmospheric Potential Oxygen (APO), to reduce the influence of terrestrial biosphere fluxes on the observed combined O2 and CO2 variations. It is found that, with the current sparse measurement network, trends in fossil fuel CO2 are derived with unrealistic interannual variations. However, a synthetic data study suggests that more robust estimates may be possible with upgraded ICOS measurements that are now coming online.*

*I find the paper to be very well researched and written, and I think it represents a novel and interesting contribution to this important topic. I have only relatively minor comments, which I hope the authors will consider.*

Thank you for your encouraging evaluation.

*General comment:*

*Another preprint is also under review on this topic involving several of the authors on this paper (Chawner et al., 2023). The two papers seem to be highly complementary, examining somewhat different aspects of the problem (indeed, the journal may feel they benefit from being published simultaneously). The findings from Chawner et al. regarding the potential influence of ocean fluxes are discussed in Section 3.2 of this paper, but I wonder if they also warrant mentioning in the Discussion too. In particular, Chawner et al. seem to find a potentially substantial, but highly uncertain and heterogeneous influence of ocean fluxes on the observations. If this is true, could they be at least partly responsible for the variability in the fossil fuel CO2 emissions derived here,*

Yes we also think so, as mentioned in line 224.

*and would they potentially reduce the scope for inferring European fluxes in future? The authors strike a positive note for the future of APO monitoring, given an expanded network (e.g., first line of the discussion). But could the value of these new measurements be somewhat compromised by our ability to accurately model, or observe with sufficient measurement density, these ocean contributions? I know that these questions are not possible to answer here, but the authors may feel that it is worth adding this context to the Discussion.*

We agree that this point had indeed not been made clear enough.

The inversion infers the fluxes from spatial gradients (and changes) in the atmospheric abundance. In a situation with only few stations, the spatial gradients are far-reaching and thus linked to fluxes from both land and ocean. Thus, the inversion cannot well separate land and ocean. In forward simulations (such as done in Chawner et al., 2023) this manifests itself by a strong dependence of the simulated APO values (more precisely, the APO enhancement with respect to the boundary conditions) on the ocean fluxes used.

With more and more stations, however, the inversion is more and more able to use the gradients between continental stations that do not depend so strongly on the ocean fluxes. Thus, land flux estimates get more independent on the ocean fluxes, which is indeed seen in the presented synthetic inversions (it would be more difficult to show this effect in a forward simulation – maybe it would be visible in gradients over the continent). Therefore, we feel that our more optimistic view is justified.

As we did not find a good way to add these explanations into the Discussion section without breaking its line of thought, we added the following sentence to the end of Sect. 3.2 where Chawner et al. (2023) is being discussed: "As confirmed by the expansion of the well-constrained area in-between the stations in Fig 5, the availability of more and more continental stations alleviates this dependence on the ocean flux, because the inversion can more and more rely on APO gradients within the continent being less influenced by the ocean fluxes."

To explain the context even better, we added an additional piece of information into the middle of the same paragraph: "Further, in test inversions where the ocean fluxes are fixed, the FF fluxes can be reconstructed more easily (test not shown, note that such inversions are unrealistic as they pretend the ocean flux to be known)."

*Minor comments:*

*Line 83 and 85: Very nit-picking, but atmospheric inversions do not necessarily require a quadratic cost function ("As in any inversion calculation").*

We agree and changed into "As mostly done in atmospheric inversion calculations..."

*Line 106: Missing "the" in "the yearly time scale"*

Added

*Line 107: Missing spaces after E and N.*

Added

*Section 2.3: It'd be useful to briefly introduce the different measurement sets here (e.g., Set A, B, etc.).*

Good point, we added paranthesized "referred to as Set A" and "Set B".

*Line 122: Perhaps "well-mixed boundary layer conditions", rather than "atmospheric conditions", which seems too broad.*

We changed into "as the atmospheric boundary layer is well-mixed".

*Like 123: You may wish to use another set of verbs than "selected or adjusted" here, which some may cause some readers to worry. The appendix describes some data filtering or correction/calibration factors being applied, which is entirely appropriate.*

We changed into "In order to jointly use the APO data from different laboratories in the inverse calculation, and to account for some measurement issues, some data selection or adjustment was necessary, as motivated and described in App B."

*Line 190 - 191: This sentence seems very strong, especially given that the potential caveats expressed in my general comments regarding ocean fluxes, which aren't explicitly included in the investigation here.*

We replaced "is" by "should be", to account for obstacles not thought of here. Otherwise, given the prospect from more stations as discussed above and as the sentence refers to "fundamental obstacles" only, we feel that it describes the situation more or less appropriately.

*Line 343 and 344: Missing spaces after the "degree" symbols.*

Added.

*Line 354: I'm not sure what "sweepingly" means here? Do you mean that the 0.4ppm is applied to all observations?*

Yes exactly. As we realize that the unclear term "sweepingly" is not actually needed, we removed it, and slightly reformulated into a separate sentence, "To this assumed model uncertainty, an assumed measurement uncertainty of 0.4ppm is added quadratically."

*Line 371: Missing spaces after E, N and degree symbols.*

Added.

*Line 378: I wasn't sure what "We also take over the result" means.*

We replaced by "We do not perform a new "$n\sigma$" outlier detection (Rödenbeck et al., 2018) for the regional run, but use the data as selected by the "$n\sigma$" outlier detection from the global runs"

*Line 399: What does a "rectangular-shaped enhancement" mean? Do you mean that there are two step changes (one up and one down)?*

Yes, exactly. We added the explanation "(step-like changes up and back)" in paranthesis.

*Figure 8 caption: Missing space after "2020-06-04".*

Added.

**Reply to Reviewer comment RC2**

We thank the reviewer for commenting on our manuscript. In the following, the original reviewer comments are given in italics.

*This is an inversion study that uses measurements of O2/N2 ratio and CO2 mole fraction (APO) to constrain fossil-fuel CO2 emissions in western Europe. The research focused on a yearly time scale spanning the period from 2007 to 2021, with the aim of establishing constraints on its decadal trend. While the current sparse APO observations appeared insufficient to provide reliable information, the authors carefully estimated uncertainties and demonstrated the method's potential for their purpose, especially as additional APO measurements become available in the future. The manuscript is well written, cautious in its conclusions, and comprehensive in its discussions. I only have some minor comments which should be better stressed before its acceptance for final publication.*

Thank you for your encouraging evaluation.

*1. Line 222-225, what is meant by "anti-correlations between land and ocean fluxes"? Could the authors consider displaying the relevant figure to better illustrate this, possibly placing it in the supplement.*

"a-posteriori anticorrelations" refer to the a-posteriori covariance matrix of the Bayesian inversion. In order to use the terminology fully correctly, we added "the errors of".

In principle, the a-posteriori covariances can indeed be calculated and presented. However, as the full matrix is huge, this would require the choice of specific functionals of the flux (aggregations) for which covariances are calculated and shown. While we could use the Western European integral and say a yearly average for the land flux, it is much less obvious what region to choose for the ocean flux. Even then, the actual calculation is relatively expensive. On the other hand, the indirect evidence for the land-ocean anticorrelations is already quite strong. Given this, we are not convinced that an additional figure showing anticorrelations would be warranted. However, in addition to the evidence already given, we added information on further evidence not mentioned so far: "Further, in test inversions where the ocean fluxes are fixed, the FF fluxes can be reconstructed more easily (test not shown, note that such inversions are unrealistic as they pretend the ocean flux to be known). Both these findings illustrate the presence of a-posteriori anti-correlations...."

*2. As mentioned in the text, transport model errors could be one of the factors contributing to the large interannual variations in figure 3, but this aspect was less discussed. The authors may want to consider discussing how transport errors could potentially affect their results.*

We added the following at the end of Sect 3.1: "Transport model errors are generally found to play a substantial role based on various intercomparisons in the literature (e.g.,Monteil et al., 2020; Munassar et al., 2023). Indeed, the results of the global inversion (App A1, results not shown) differ considerably from those of the regional inversion (App A2, Fig 3), their difference mainly being the transport model."